# The Impact of Ecological Restoration on Biogeochemical Cycling and Mercury Mobilization in Anoxic Conditions on Former Mining Sites in French Guiana

**DOI:** 10.3390/microorganisms9081702

**Published:** 2021-08-10

**Authors:** Ewan Couic, Alicia Tribondeau, Vanessa Alphonse, Alexandre Livet, Michel Grimaldi, Noureddine Bousserrhine

**Affiliations:** 1Observatory of Sciences of the Universe, UMR 6118 Géosciences, 35000 Rennes, France; 2Water, Environment and Urban Systems Laboratory (Leesu), University of Paris-Créteil, 94010 Créteil, France; vanessa.alphonse@u-pec.fr (V.A.); livet@u-pec.fr (A.L.); 3National Museum of Natural History, 75005 Paris, France; alicia.tribondeau@edu.mnhn.fr; 4Development Research Institute, 93143 Bondy, France; michel.grimaldi@ird.fr

**Keywords:** biogeochemical cycle, iron-reducing bacteria, sulfate-reducing bacteria, mercury solubilization, ecological restoration, French Guiana

## Abstract

Successive years of gold mining in French Guiana has resulted in soil degradation and deforestation leading to the pollution and erosion of mining plots. Due to erosion and topography, gold panning sites are submitted to hydromorphy during rainfall and groundwater increases. This original study focused on characterizing the impact of hydromorphic anaerobic periods on bio-geochemical cycles. We sampled soil from five rehabilitated sites in French Guiana, including sites with herbaceous vegetation and sites restored with fabaceous plants, *Clitoria racemosa* (Cli) mon-oculture, *Acacia mangium* (Aca) monoculture, *Clitoria racemosa* and *Acacia mangium* (Mix) bi-culture. We conducted mesocosm experiments where soil samples were incubated in anaerobic conditions for 35 days. To evaluate the effect of anaerobic conditions on biogeochemical cycles, we measured the following parameters related to iron-reducing bacteria and sulfate-reducing bacteria metabolism throughout the experiment: CO_2_ release, carbon dissolution, sulphide production and sulphate mobilization. We also monitored the solubilization of iron oxyhydroxides, manganese oxides, aluminum oxides and mercury in the culture medium. Iron-reducing bacteria (IRB) and sulfate-reducing bacteria (SRB) are described as the major players in the dynamics of iron, sulfur and metal elements including mercury in tropical environments. The results revealed two trends in these rehabilitated sites. In the Aca and Mix sites, bacterial iron-reducing activity coupled with manganese solubilization was detected with no mercury solubilization. In herbaceous sites, a low anaerobic activity coupled with sulphide production and mercury solubilization were detected. These results are the first that report the presence and activity of iron- and sulfate-reductive communities at rehabilitated mining sites and their interactions with the dynamics of metallic elements and mercury. These results report, however, the positive impact of ecological restoration of mining sites in French Guiana by reducing IRB and SRB activities, the potential mobility of mercury and its risk of transfer and methylation.

## 1. Introduction

In the Amazon rainforest, several decades of gold mining and other anthropogenic activities has led to major environmental disruptions due to the cumulative effects of deforestation, acid mine drainage and soil and water pollution from arsenic, cyanide and mercury contamination [1]. Mining activity in the Amazon also leads to substantial degradation of the soil and its organic matter content [2], disruption of soil microorganism communities and potential prevention of the expected ecological succession [3,4].

Recent mining operations modify the topography of mining lands and expose the soils to sunlight and inclement weather conditions, and the absence of vegetation cover can facilitate erosion and the transport of particles containing trace elements into the water network [5]. In addition, this exploitation can create disorganized hydromorphic zones where anaerobic conditions can favor the activity of facultative and strict anaerobic bacterial communities [6].

Soil microorganisms play an important role in the fate of metals in soils. Certain autotrophic bacteria of the genus *Acidithiobacillus* in aerobic environments can oxidize metal sulphides [7,8], thus acidifying the environment due to sulfuric acid produced as consequences of their metabolism, and they can also reduce iron and manganese [9,10], which promotes the release of some oxide adsorbed metals [11]. Under anaerobic conditions, the genus *Desulfovibrio* can reduce iron-hydroxysulfate minerals to sulphides [12] and affect the mobilization of their associated trace elements. The sulphides released in this way can, in turn, be associated to soluble trace elements, such as mercury, and form insoluble metal sulphides, such as cinnabar HgS. Sulfate-reducing bacteria (SRB) are also known as important methylators of inorganic Hg in anaerobic environments [13,14]. Under anaerobic conditions, iron-reducing bacteria (IRB), such as *Geobacter metallireducens*, *Clostridium sporogenes* and *Clostridium butyricum*, solubilize iron oxides by reducing Fe(III) to Fe(II), using it either as a major final electron acceptor in anaerobic respiration or as a minor electron acceptor during fermentation metabolism [9,10,15]. The solubilization of iron oxides is concomitant with the solubilization of its associated trace metallic elements. Indeed, it is now recognized that the iron biogeochemical cycle is associated with trace elements mobility such as mercury [16,17]. Recent studies suggest that iron-reducing bacteria have also been involved in the methylation of mercury. In this context, iron-reducing bacteria (IRB) were described as major agents that solubilize iron oxyhydroxide in anaerobic conditions and release the associated toxic metal [18,19,20,21,22]. Several studies have demonstrated the presence and activity of iron-reducing bacteria (IRB) and sulfate-reducing bacteria (SRB) in natural sites and in gold mining sites in French Guiana [6,18,23].

In soils, mercury mobility modelling is complex, and its retention on clay minerals, on organic matter and on Fe-, Al- and Mn-(oxyhydr)oxides depends mainly on pH, redox conditions and the activity of anaerobic microbial communities [24], which in turn depend on the topographic position, vegetation cover, hydromorphy and temperature [25,26]. While there is no doubt that iron-reducing and sulphate-reducing bacterial activities play a major role in the mobility of trace metal in natural tropical soils, no study to our knowledge has investigated these activities in restored soils to make it a means of evaluating the performance of the restoration. Only a few authors were interested in these mechanisms on Guyanese mining sites, and no study deals with microbial activity under anaerobic conditions on rehabilitated and restored former mining sites.

Due to the contributions of legislation in the management of gold mining sites, mining companies are now required to rehabilitate mining sites and, if possible, carry out an ecological restoration protocol [27]. Legume species have been used in restoration efforts mainly because of their ability to grow on poor substrates while renewing the organic matter stock and limiting erosion [28]. However, this legislation is recent, and there is therefore a lack of data concerning the impact of ecological restoration in French Guiana on the fate of metallic elements, particularly under anoxic conditions.

The objective of this work is to evaluate the impact of different protocols of ecological restoration of mining soils on the mobility of the trace elements through the evaluation of two major bacterial activity in these soils, which are the iron-reductive activity and sulphate-reducing activity. Theses are recognized as the actors controlling not only the dynamics of iron in these soils but also that of toxic trace metal elements, including mercury, associated with iron, organic matter and other soil components. Soil samples from rehabilitated and restored plots with different fabaceous species (*Clitoria racemosa* and *Acacia mangium*) and herbaceous species (*Lycopodiella sp*, *carex sp*) were incubated under anaerobic conditions. The release of metals from soils into the culture medium and measurements of anaerobic microbial metabolism were monitored throughout the incubation period. This approach improves our knowledge of microbial communities at rehabilitated sites in Guyana and our ability to assess the effect of this rehabilitation on soil quality.

## 2. Materials and Methods

### 2.1. Site Description and Soil Sampling

The work was undertaken in French Guiana, South America. Annual rainfall ranges from 2 m to 4 m, and the annual average temperature is approximately 26 °C. The climate is classified as seasonal equatorial [29]. The soils sampled for this study were rehabilitated after the more recent gold mining extraction, which occurred between 1990 and 1997. After closing the alluvial mine pit, rehabilitation consisted of returning the original soil that was excavated during mining. All excavated soils were then homogenized and flattened before ecological restoration began [28]. These soils could be considered anthroposols, and they have different structures and textures than natural tropical forest soils in the same topographic position. The five sites chosen to represent the different restoration modality adopted were already described in a previous research article [30] and are illustrated in Figure 1.

Briefly, the first site, in the region of Yaoni, is only covered from 5% to 10% of the soil surface and is only dominated by herbaceous species, and this site has not been restored. This site was named NR to denote “Non-Restored”. The second selected site in the region of Belizon was rehabilitated, and more than 75% of the site is now mostly covered by *Lycopodiella sp*. and some *Cyperaceae* species (*Cyperus sp* and *Carex sp*), and no trees are present. This site was named Lyc to denote “Lycopodiella”. The third restoration site, in the region of Belizon, was restored by *Clitoria racemosa* in monoculture. The trees were planted with macro-cuttings in January 2013. This site was named Cli to denote “Clitoria”. The fourth restoration site, in the region of Yaoni, was restored by *Acacia mangium* in monoculture. The trees were planted with macro-cuttings in 1998. This site was named Aca to denote “Acacia”. The fifth restoration site, in the region of Yaoni, was restored by *Acacia mangium* and *Clitoria racemosa* (N 04°29.595′/W 052°26.984′). This site was named Mix to denote “Mixed fabaceous”. The trees were planted with macro-cuttings in 1998. 

The sampling campaign took place between April and June 2016 in French Guyana. This period usually corresponds to the main Guyanese rainy season, and the ambient humidity felt during the sampling was very high (not measured) with temperatures between 27 and 30 degrees during the sampling. At each site, soil samples were collected from depths between 0 and 10 cm with an auger. The sampling at the 5 sites consisted of sampling 5 plots (*n* = 5) per site, with 3 sub-samples per plot that were pooled at the laboratory. The 5 sampling plots per site were selected based on the topography of the terrain, the accessibility of the plots and the feasibility of sampling. All plots were in similar topographic environments at similar altitudes. Soil samples were immediately sealed in sterile hermetic polyethylene bags for transportation to the Cayenne IRD laboratory. Samples were then air-dried at ambient temperature (25 °C) for approximately 3 weeks. These 75 collected samples (5 sites × 5 plots × 3 samples per plot) were then sieved at 2 mm to remove rocks, homogenized and hermetically sealed at 4 °C until use.

### 2.2. Initial Soil Sample Characterization

Before the soil microcosm studies, pH, total major element (i.e., C, N, P, Fe, Al, Mn) content, total trace element (Hg) content, total organic carbon content, granulometry and microbial biomass carbon (using the substrate-induced respiration (SIR) method) were determined to characterize the main physicochemical and biological properties; these measurements enabled us to assess the main differences between soils at initial conditions.

#### 2.2.1. Soil Total Major Element Content

For all metal analyses, the samples were first ground to 63 µm. To obtain total metal (i.e., Fe, Al, Mn) concentrations, soil samples were dissolved in cleaned Teflon digestion tubes (SCP Science) at 80 °C in a mixture of concentrated HNO_3_-HF (3:1 mL) using a hot block (Digiprep MS, SCP Science) following the ISO 11466 principles with some modifications advised by the device supplier. Samples (300 g) were warmed at 80 °C. Samples (300 mg) were then dissolved in 2 mL of concentrated HCl overnight at ambient temperature and then warmed for 4 h at 80 °C. Then, samples were diluted in 5% HNO_3_ before analysis by ICP-OES (Spectroblue).

#### 2.2.2. Soil Total Mercury Content

Mercury concentrations (HgT) in soil samples and in the culture medium were determined directly by thermal decomposition atomic absorption spectrometry after gold amalgamation using an automatic mercury analyzer (AMA 254, Symalab) with a detection limit of 0.001 ng using the same method as described in a previous research article [30].

#### 2.2.3. Soil Carbon, Nitrogen and Phosphorus Measurements

Soil total carbon (Ctot) and total nitrogen (Ntot) were determined by the Dumas method (NF ISO 13878). Briefly, the Dumas method consists of flash combustion of the soil sample (at 1000 °C) using helium and oxygen. After purification of the combustion gases, determination of total carbon and nitrogen contents was performed by chromatography with a thermal conductivity detector (NA 1500 series 2 CARLO-ERBA).

Total phosphorus (Ptot) was determined after the acid digestion of soil samples, as previously described. The HPO_4_ formed during acid digestion was measured after adding ammonium molybdate and ascorbic acid [31] using the recommendations of the Spectroquant^®^ phosphate test (ref: 114848). The color produced by phosphomolybdenum (PMB) was measured by colorimetry at 885 nm (Genesys 10 µv Scanning, Thermo).

#### 2.2.4. Total Organic Carbon in Soil Samples

Water extractions were performed for total organic carbon (TOC), and concentrations were measured with a Shimadzu TOC-500 apparatus (Shimadzu, Kyoto, Japan). Briefly, 1 g of soil sieved at 2 mm was shaken for 24 h in a polypropylene centrifuge tube with 10 mL of ultrapure water. The suspensions were centrifuged at 2000 rpm for 10 min, and supernatants were filtered at 0.2 µm (PTFE, VWR©) and directly analyzed on the TOC analyzer.

### 2.3. Experimental Conditions

Microcosms were prepared to study microbial activity under anaerobic conditions. Thus, 125 mL sterile hermetic glass plasma bottles were filled with 15 g of soil and 100 mL sterile ultra-pure water. Microcosms were performed with 3 replicates for each plot at each site for a total of 75 microcosms (5 sites × 5 plots per site × 3 microcosms per plot).

The microcosms obtained were mixed with sterile spatulas and sealed with a butyl/PTFE shield cap (VWR©). Anoxic conditions were then initiated by flushing microcosms with N_2_ for 10 min. The microcosms were incubated in the dark at 28 °C for 35 days without shaking. The measurements were taken on days 7, 14, 21, 28 and 35. Then, 8 mL of medium culture was collected with a sterile syringe through the butyl/PTFE shield cap, which was then analyzed for pH and the following concentrations in solution: sulfates, sulfides, ferrous iron, total iron, aluminum, manganese and mercury. The preparation of the experimental set-up is described in Figure 2 below.

#### 2.3.1. Bacterial Metabolism

Immediately before sampling the culture medium, carbon mineralization was quantified in the microcosm atmosphere by measuring CO_2_ evolution using micro gas chromatography (micro-GC 490, Agilent).

Dissolved organic carbon (DOC) concentrations in the culture medium were measured with a Shimadzu TOC-500 apparatus (Shimadzu, Kyoto, Japan) after being filtered at 0.2 µm (PTFE, VWR©).

Sulphur oxidation was followed by measuring sulfate (SO_4_^2−^) formation in the culture medium. This was carried out using a kit (Spectroquant, Merck), similar to NF ISO 11048. This kit measures sulfate concentrations in the range of 5–250 mg SO_4_^2−^/L. Samples were filtered (0.45 µm, Minisart, Sartorius) prior to analysis.

Sulfate reduction was followed by measuring sulphide (S^2−^) formation in the culture medium. This was carried out using a kit (Spectroquant, Merck), similar to NF ISO 10530. This kit measures sulphide concentrations in the range of 0.02–1.5 mg S^2−^/L. Samples were filtered (0.45 µm, Minisart, Sartorius) prior to analysis.

Soil pH was measured in the culture medium by using a pH meter (MetrOhm 744).

#### 2.3.2. Iron Reduction in Culture Medium

To avoid any oxidation reactions, all samples used for the determination of iron (II) were immediately acidified to pH 1 with nitric acid (65%, Normatom) and stored at 4 °C before analysis. Ferrous iron (Fe(II)) was measured in the medium using a colorimetric method with orthophenanthroline-chlorydrate [32]. The absorbance was read at 490 nm (Genesys 10 µv Scanning, Thermo). Readings were compared to a standard curve established with dissolved anhydrous ferrous sulfate (Fe(NH_4_)_2_(SO_4_)_2_·6H_2_O).

#### 2.3.3. Total Iron, Aluminum and Manganese in the Culture Medium

Metals were analyzed after filtration (0.45 µm acid-washed membrane filter, Minisart SRP 25, Sartorius) and acidification at pH 1 with HCl (37%) by ICP-OES (Spectroblue).

### 2.4. Statistical Analysis

The normality of data distributions and equal variance between treatments were tested using the Shapiro test and Bartlett’s test, respectively. To study the effect of vegetation mode, the effect of incubation and the interactions between soils and time, two-way ANOVA was performed. Tukey’s HSD multiple comparison method was used to test differences throughout the experiment at specific measurement points. To refine the analysis of the data, repeated-measures ANOVA was also carried out to determine the “sites” effect by taking into consideration the temporal evolution of the variables during the experiment. To analyze the effect of the different restoration processes and experimental treatments on biogeochemical data (pH, CO_2_, TOC, SO_4_^2−^, S^2−^, Fe(II), Fe, Al, Mn and Hg), the results were also submitted to principal component analysis (PCA). The significance of each explanatory variable was tested using a Monte-Carlo permutation test. To calculate the correlations between the parameters measured during the experiment, Pearson correlations were performed. R software was used for all statistical analyses (R version 3.6.1 (5 July 2019)).

### 2.5. Quality Assurance and Control (QA/QC)

To avoid contamination, all materials used in this work were acid-washed twice with HNO_3_ (5%) and then rinsed several times with Milli-Q water before use. Concerning mercury assays in samples, the relative error was routinely 5% and was always under 10%. The detection limit (defined as three times the standard deviation (SD) of the blank) was 0.005 mg g^−1^. Concentrations obtained for repeated analyses of certified reference materials never exceeded the published range of concentrations (i.e., 0.128 ± 0.017 mg g^−1^ and 0.091 ± 0.009 mg g^−1^ for BCR-277R and MESS-3, respectively). Total elements and trace elements were determined by inductively coupled plasma optical emission spectrometry (ICP-OES, Spectroblue). Calibration was conducted using standard solutions for spectroscopy (EnviroMAT Ground Water, High ESH-2). The analytical errors were estimated by carrying out 3 measurements for each sample and were <10%.

## 3. Results

### 3.1. Soil Physical and Chemical Properties

The initial soil characteristics of the studied sites are summarized in Table 1.

The results for the granulometry demonstrated that the NR site was twice as clayey as other sites, with approximately 46.4% clay and 41.5% silt. In accordance with the soil triangle of texture, the Lyc site was silty with 65.8% silt on average. The Cli site was silt-clay-sand with 53.9% sand. The Aca and Mix sites had the same granulometry, with a dominance of sand over silt. Soil pH indicated that the site restored with *Acacia mangium* was more acidic than the other rehabilitated sites. Among the three sites restored with Fabaceae, the total carbon, nitrogen and phosphorus contents were always significantly the highest in Mix site. Macronutrient contents (C, N) for the Mix site were similar to the natural soil data in French Guiana [18,33,34]. Of the two restored sites (Aca and Cli), the Aca site had higher levels of macro-elements than the Cli site. The total carbon content was 14.11 g kg^−1^ at the Aca site and 7.01 g kg^−1^ at the Cli site. The total nitrogen content was 1.12 g kg^−1^ at the Aca site and 0.4 g kg^−1^ at the Cli site. The total phosphorus content was 0.37 g kg^−1^ at the Aca site and 0.17 g kg^−1^ at the Cli site. The total carbon, nitrogen and phosphorus contents at the spontaneous vegetation site NR were low. The five sites were rich in iron with a minimum of 44.56 g kg^−1^ at the Lyc site and a maximum of 113.3 g kg^−1^ at the Cli site. The aluminum contents at the sites were similar and were higher than the natural content in French Guiana. Manganese levels are low at the Lyc and Cli sites (0.10 g kg^−1^ and 0.11 g kg^−1^, respectively), and the Mix and NR sites had the highest concentration of total Mn. Mercury levels for NR, Lyc and Cli were lower than for other sites. The three sites restored with fabaceous species had significantly higher soil microbial biomass levels than the NR and Lyc sites. The maximum biomass was for the Mix site, while the minimum biomass was for the NR site, with values of 1103 mg kg^−1^ and 223 mg kg^−1^, respectively.

### 3.2. Sulfate and Sulfide Concentration in the Culture Medium

Sulfate (Figure 3A) and sulfide contents (Figure 3B) in the culture medium were both affected by vegetation cover and incubation period, and the different sites evolved differently during incubation. For the five sites, the sulfate concentration in the solution increased during kinetics, with stagnation from day 21 for all sites except the Aca site. The highest sulfate concentrations were observed at the Lyc, Aca and Mix sites, while the restored Cli site showed the lowest increase in sulfate concentration during incubation. The evolution of sulfide concentrations showed very different trends. There was no sulfide production in the NR environment and very low production in the environments of the two restored sites Aca and Mix. For the Lyc and Cli sites, a strong increase in sulfide levels was observed between day 21 and day 28 of incubation.

### 3.3. Heterotrophic Metabolism and Dissolved Carbon in the Culture Medium

Heterotrophic respiration (Figure 3C) and organic carbon dissolution (Figure 3D) were both affected by vegetation cover and incubation period, and the interaction of these two factors had a clear effect. Heterotrophic respiration for Aca and Mix were quite close and significantly higher than that at other sites. The non-restored Lyc site had a greater heterotrophic respiration than the restored Cli site and NR site. The dissolved organic carbon content in the culture media was significantly higher in the two restored sites, Aca and Mix, than in the other three sites, with a clear increase until day 21. For the NR, Lyc and Cli sites, soluble carbon production was very low, with a slight peak in the first 15 days of incubation.

### 3.4. Metals Solubilization

The ANOVA showed (Table 2) that all trace element concentrations in the culture medium were affected by both vegetation cover and incubation period, and the different sites evolved differently during incubation.

#### 3.4.1. Iron in the Culture Medium

Concentrations of both total iron (Figure 4A) and iron (II) (Figure 4B) followed similar trends. No iron solubilization was detected in culture media at NR sites throughout incubation. Total iron solubilization was detected similarly in the Lyc and Cli microcosms, but the reduction of iron to iron (II) was slightly greater for the Lyc than Cli site during the first two weeks of incubation. The evolution of total iron solution levels for the Aca and Mix sites was similar during the first three weeks, but a shift was observed from this date with an increase in iron concentration for the Mix site and stagnation for the Aca site. Concerning iron reduction, rates for the Aca and Mix sites was the same, but with approximately twice as much reduction observed for the Mix site.

#### 3.4.2. Aluminum in the Culture Medium

The evolution of soluble aluminum levels (Figure 4C) in the media looked very different from that observed for iron and manganese solubilization measurements. No aluminum levels were detected for the NR and Lyc sites during incubation. Slight aluminum solubilization was detected for the Aca site but without a significant change during incubation. However, the Cli and Mix sites had peak concentrations on day 14 for Cli and on day 21 for Mix at approximately 0.8 mg and 1.4 mg Al L^−1^ for the Cli and Mix sites, respectively.

#### 3.4.3. Manganese in the Culture Medium

Measurements of soluble manganese in solution in culture media (Figure 4D) showed very similar trends to those of total soluble iron. Indeed, manganese detection was very low for the NR, Lyc and Cli sites, whereas the Aca and Mix sites had significantly higher manganese solubilization. Manganese levels in media stabilized from day 21 for Aca and 28 for Mix at concentrations of approximately 8 and 10 mg Mn L^−1^ for Aca and Mix site samples, respectively.

#### 3.4.4. Mercury in the Culture Medium

Measurements of soluble mercury in media (Figure 4E) showed two significantly distinct trends. On the one hand, the restored sites with significant vegetation cover (Cli, Aca and Mix) had very low levels of soluble mercury, without major changes in concentrations during incubation. On the other hand, the two sites that were not restored by tree cover (NR and Lyc) showed a significant increase in soluble mercury levels that stabilized at a concentration of approximately 0.35 ng Hg L^−1^ on day 28 before decreasing at the end of incubation.

#### 3.4.5. pH in the Culture Medium

The ANOVA showed that medium pH (Figure 4F) was affected by vegetation cover (*p* < 0.05), but repeated-measures ANOVA showed that the pH change during incubation was similar between different soils, with no incubation time effect. The two sites without tree cover (NR and Lyc) had significantly more acidic pH levels compared to the restored sites (Cli, Aca and Mix sites). About trends, the Mix site had a similar pH variation of NR and Lyc sites in the first week of incubation, but after this time the pH behavior of the Mix site was similar to those in Cli and Aca sites.

### 3.5. Principal Component Analysis of Microbial Reducing Activities and Metal Solubilization

*t*-test comparisons enabled us to distinguish significant differences between the five sites with clusters (*p* = 0.001). The data clusters for the NR and Lyc sites were not significantly different from one another but were very different from the three clusters for the restored Cli, Aca and Mix sites. The cluster for the restored Cli site seemed to be relatively separate from the other restored sites, whereas the Aca and Mix sites were very close. In this study on the impact of the nature of the vegetation cover, the quality of ecological restoration was detected (Figure 5a), as shown in Figure 5, which shows the results of PCA analysis on the data set obtained after incubation under a controlled anaerobic conditions approach. The Eigen values calculated by the PCA statistical analysis determined that two axes were significant (Figure 5b), accounting for 54.4 and 17.5% of the total variance. Axis 1 was best represented by variables consisting predominantly of heterotrophic respiration and soluble content of dissolved carbon, sulfates, iron and aluminum, while Axis 2 was mainly linked to sulfide concentration in solution. In addition, mercury concentration and pH were negatively correlated.

## 4. Discussion

The main objective of this study was to determine the impact of the anaerobic microbial activity on metal solubilization in soils of old gold mines rehabilitated and restored by revegetation protocols. A kinetic approach under controlled reducing conditions was adopted in which factors related to carbon and sulphur metabolism were measured. The solubilization of iron, aluminum, manganese and mercury in culture media was also monitored to link the impact of different ecological restoration approaches on the mobilization of certain metals.

In iron-rich soils such as Guyanese mining soils, it is generally accepted that iron molecules, and in particular iron oxides, are sinks for a wide range of metal elements, including aluminum and manganese, and trace elements, such as mercury or cadmium, and that the biogeochemical cycles of these elements are closely linked [10,17,19,20,35,36]. Mercury has also been shown to have affinity for Al- and Mn-(oxyhydr) oxides [17,24,37], particularly in oxide-rich soils such as Guyanese soils. However, bacterial solubilization of these oxides can lead to re-mobilization of mercury and increase its mobility and toxicity [18,19,20,21,22]. While these microbial mechanisms have been described in natural or anthropized soils, no studies have addressed these processes in rehabilitated soils. These results allow us to estimate the influence of different ecological restoration protocols on reducing microbial activities and on the potential mobility of mercury on rehabilitated sites, which are essential data needed to assess the quality of ecological restoration.

### 4.1. Properties of the Rehabilitated Soils and Distribution of the Various Trace Elements

#### 4.1.1. Soil Texture of Rehabilitated Soils

Although the concentrations of oxides, clay, silt and sand in the soils were close to those for natural Oxisols or hydromorphic soil encountered in French Guyana [18,38], the soils in this study were all completely destructured and redesigned due to gold mining. Therefore, they can be considered anthroposols, and the comparison of their soil properties with natural Guyanese soils such as oxisols, acrisols or Gleysols does not seem relevant. The differences in soil texture were primarily due to uneven site reworking after gold mining. These differences were observed not only at the Belizon sites, between the silty Lyc site and the sandy Cli site, but also at the Yaoni sites, between the silty clay NR site and the two sandy-silty Aca and Mix sites. Such variation in texture for nearby sites that had the same mining process was noted in Schimann’s (2007) work on rehabilitated sites in Guiana [39] and show difficulty in applying the recommendations of ecological restoration management [28] and the instructions of the mining code [27]. If the significant textural differences make it difficult to compare the pedogenesis of these soils, then this study highlights the consequences of this heterogeneity; regardless, the comparison of sites seems interesting. Due to the mobility of metals in tropical soils being linked to the physical properties of the soil, there could also be an impact on metal transfers in the water network. Variations in metal contents must be taken into consideration. Total iron concentrations in the sample soils were similar to concentrations found in gleysols and oxisols [18], with strong heterogeneity between sites but no significant correlation with soil texture parameters, carbon content (*p* = 0.62) or microbial biomass (*p* = 0.07) and no relationship to tree covering. The forms of iron in these soils could be different, which would impact solubilization kinetics. The distribution of manganese and aluminum also appeared to be heterogeneous among the various rehabilitated sites. 

#### 4.1.2. Mercury Content of Rehabilitated Soils

Mercury concentrations in these restored soils were still low when compared with other reported gold-mined soils in French Guiana, remaining within the normal range of Hg in this specific area [34]. Nevertheless, mercury levels may reflect different geochemical behaviors. Mercury concentrations were positively correlated with organic matter concentration (*p* = 0.0001) and microbial biomass (*p* = 0.0001), which is consistent with the literature, as described by Schuster et al. (1991) and Skyllberg et al. (2000), showing the high affinity of mercury for organic matter, particularly for sulphide groups [40,41]. In addition, the dense vegetation cover from the Aca and Mix sites may have contributed to fixing mercury in soils and limiting its mobility [42]. Vegetation allows atmospheric mercury to be fixed in leaves and in litter deposits. This result could confirm the positive contribution of ecological restoration in limiting mercury transfers into the water network by limiting erosion processes; however, no samples could be taken directly after ecological restoration, and the heterogeneity of initial mercury concentrations cannot be excluded.

### 4.2. Microbial Activities during the Anoxic Soil Experiment

Firstly, there were significant differences in soil microbial biomass, which were related with the density and quality of the vegetation cover on the restored sites. These results, already highlighted by various works on rehabilitated sites, confirm the contribution of successful ecological restoration in the recovery of soil microbial biomass [39,42,43,44], especially on the rehabilitated sites of Yaoni.

The experimental variables measured during the anaerobic experiment indicated different microbial behaviors under reducing conditions depending on the ecological restoration modalities. The results were confirmed by the significant (*p* < 0.001) results of the PCA (Figure 5), which showed a clear separation of the rehabilitated sites according to the nature of their vegetation cover. Thus, the two restored sites, Aca and Mix, were on the positive side of axis 1 and were mainly characterized by important biological and reducing activities (Fe and Mn), with the Cli site relatively close but on the negative side of axis 1. These three sites, where ecological restoration has been accomplished with the efficient establishment of fabaceous macro-cutting, are different from the two non-restored sites NR and Lyc, which were mainly characterized by a strong dissolution of mercury and are sites where ecological restoration has not been controlled and effective. Except for mercury, the soil respiration was strongly correlated with microbial biomass concentrations in the samples, and no dissolution of iron and aluminum could be detected in the media containing the NR soil samples.

#### 4.2.1. Heterotrophic Activities and Iron Reduction

During the experiment, CO_2_ production, dissolved organic carbon and sulfate solubilization were significantly correlated with iron and manganese dissolution (Table 3), which would indicate that the Fe and Mn reduction is of biological origin [10,45]. Differences in dissolved organic carbon concentrations between restored and non-restored sites could be explained by differences in total microbial biomass and therefore total activity, as well as the quality and dynamics of organic matter. It would, therefore, be interesting to characterize the labile and recalcitrant organic matter as suggested by Schimann (2007) [39]. From a thermodynamic point of view, iron reduction is generally more efficient under reducing conditions [9] and occurs before sulfate reduction and CO_2_ production through methanogenesis by acetotrophic and methylotrophic Archaea [46], which seems to agree with the results observed in this study, but this hypothesis cannot be validated with results. These results appeared to confirm the presence and activity of iron-reducing bacteria at the restored Aca and Mix sites. For the Aca and Mix sites, the results show an interesting coupling between the dissolution of iron and manganese oxides, probably in the form of Mn^2+^ due to the maintenance of reducing conditions throughout the experiment. The strong correlation between manganese and total dissolved iron (R^2^ = 0.977), also observed by Gounou et al. (2010) [21] and Bousserrhine et al. (1999) [10], could be explained both by the presence of manganese-reducing bacteria or by the release of adsorbed manganese to iron oxides when ferric iron is reduced [10,21,47]. It is interesting to note the differences between the solubilization of iron and Mn and the solubilization of aluminum. The dissolution dynamics of aluminum oxides could indicate different speciation of metal elements between sites. The rapid dissolution of aluminum oxides for the Cli site could indicate the presence of amorphous oxides, which are easily soluble, while in the Aca site, the very low mobilization of aluminum would indicate the presence of crystallized oxides. For the Mix site, the later dissolution of aluminum relative to iron could indicate the presence of substituted iron oxides containing aluminum, which would therefore be solubilized more slowly. These differences highlight the importance of oxide species and character in metal mobilization mechanisms, and it has been shown that the nature of oxides is one of the most important factors involved in metal element mobilization processes [10,18].

#### 4.2.2. Sulfate-Reducing Activities

Iron-reducing activities were also supported by the rapid increase in sulfate levels in culture media, which may be associated with desorption of sulfates bound or adsorbed to Fe(III)-(hydr)oxides [48]. Although it was demonstrated that sulfate-reducing bacteria are also involved in the reduction of iron oxides [9,49], their presence in these rehabilitated sites did not seem to be obvious. Indeed, there is no correlation between sulfate and sulphide content experimental variables. There is also no positive correlation between sulfate concentrations and sulphide concentrations, which would indicate very different sulfate-reducing activities between the rehabilitated sites. The peak in sulphide concentration for the Lyc and Cli sites would confirm the hypothesis of the presence and activity of SRB, but the results for the other sites would indicate heterogeneity of SRB communities in the rehabilitated soils.

The low sulphide contents during the experiment in NR, Aca and Mix samples suggest that sulfate-reducing communities are not very active, and several factors could explain these low activities. In the Aca and Mix sites, the low SRB activities could be explained by high competition with iron-reducing bacteria, generally described as more competitive in an acidic environment [50,51,52]. For the NR site, the more acidic pH during the experiment could also explain the low sulfate-reducing activities because these bacteria are generally more active in a pH range of 6–8 [53,54]. The very low overall heterotrophic metabolic activities recorded for the NR site could also be the cause of this lack of sulfate-reducing activities.

However, these chemical measurements may not reflect the observations of the culture media during the experiment. After 15 days of incubation, all the culture bottles had turned blackish, and the culture medium samples had a very marked smell of H_2_S, indicating a priori the presence of iron sulphide, certainly linked to the presence of SRB.

The very low sulphide contents measured could then be explained by both H_2_S degassing and potential metal sulphide precipitation [55]. Indeed, the possible precipitation mechanisms of H_2_S with iron under acidic conditions could explain the very low levels measured during the experiment [56,57]. Coleman demonstrated that sulfate-reducing bacteria could reduce iron enzymatically [58], the presence of sulfides in culture media could correlate with Lovley’s work, where he associated iron reduction and iron sulfide production [9]. Thus, in this experiment, the activities observed correspond to the interactions between reducing communities. It would be interesting to test this hypothesis by discriminating between the two communities with the addition of specific substrates such as acetate or lactate to favor sulfate-reducing communities.

### 4.3. Mercury Dissolution during the Anoxic Soil Experiment

The dissolution of mercury in mesocosms under anaerobic conditions showed very contrasting results with other metallic elements, and the principal component analysis (Figure 5) clearly showed dissolution of mercury, which depends on the density of the vegetation cover, associated with an increase in solubilization processes on the least restored NR and Lyc sites. No significant correlation was found (Table 3) with the other controlled variables during the experiment, except for a negative correlation with pH (R^2^ = −0.621). These results confirmed that soil pH is a key factor affecting the adsorption–desorption behaviors and, hence, the bioavailability of heavy metals in soil, especially mercury [59]. In the literature, iron oxide reduction processes, mainly microbial [60], are at the origin of the release and mobilization of adsorbed trace elements [9,10,19,20], including mercury. However, anoxic soil conditions are not always synonymous with the mobilization of mercury [61]. In this study, the reduction and dissolution of iron, manganese and aluminum, which are considered important carrier phases of mercury, have a relatively limited effect on mercury mobilization in the culture medium. These results could be similar to Harris’s (2011) work [18], which showed that bacterial reduction of iron oxides, unlike chemical reduction, was not always synonymous with mercury mobilization. However, it cannot be excluded that mercury speciation in soil phases may have changed during the experiment, a result also highlighted by Harris (2011) [18]. Although the dissolution of mercury and the other variables controlled during the experiment are not correlated, negative correlations of mercury dissolution were measured with the parameters related to the organic matter content in the soil. For microbial biomass, TOC, carbon and total nitrogen, the correlations with mercury dissolution were r = −0.87, r = −0.55, r = −0.5 and r = −0.51, respectively. These results could mean that the higher the organic matter content in reclaimed soils, the greater the retention of mercury and the lower its mobility. However, the measurement of total microbial biomass does not only reflect the presence of IRBs, SRBs and methanogens, nor do quantitative measurements of organic matter reflect its nature and quality. Several authors have highlighted similar results. Anderson [37] showed that the affinity for and retention of mercury in organic matter in acidic soils could exceed the affinity for metal oxides, as confirmed by recent studies on mercury speciation at former Guyanese mining sites [34,42]. Roulet and Lucotte’s [16] work highlighted that mercury retention in Guyanese soils was not always correlated with the presence of metal oxides. However, recent studies show that the retention of mercury, if it is adsorbed on iron oxides, would depend more on the nature of the oxides than on their total concentration in the soil. Harris et al. (2009) and Guedron et al. (2009) have indeed shown on Guyanese sites that iron speciation and the amount of crystalline and amorphous oxides are important drivers of mercury dynamics. Differences in mercury speciation in soil phases could therefore be a key factor in its solubilization and mobilization [62]. Speciation of mercury at the restored Cli, Aca and Mix sites, due to microbial dynamics and organic material turnover approaching a natural site, may have been altered to insoluble forms such as HgS. Indeed, due to its chalcophile character, mercury in natural conditions in well-oxygenated soils is mainly found in the form of cinnabar (HgS), a sulfide that is not easily weathered [62]. The increased presence of soluble and non-adsorbed forms of mercury at NR (Figure 4) and Lyc sites, including alkyl and inorganic mercury species, may also confirm the solubility and reactivity of mercury at rehabilitated but non-restored sites is greater than that at restored sites [42]. In this study, soluble mercury is also negatively correlated with total mercury (r = −0.68), which could show that mercury mobility is not related to its total concentration in soil. The low concentrations of dissolved mercury at the Cli, Aca and Mix sites could also be explained by the high presence of sulfate-reducing bacteria. Under anoxic conditions, these bacteria will produce sulphides that can complex with mercury to form mercuric sulphide complexes [63], which is very insoluble compared to other forms of mercury [62], and form the black precipitate in the mesocosms that was observed in this experiment and in the Gounou experiment (2012) [21]. Although the dissolution of HgS is possible in acidic media [62,64] with low molecular mass organic acids and root exudates, the experiment may not have lasted long enough to observe significant changes in dissolved mercury in the restored Cli, Aca and Mix sites. In addition, the estimation of other forms of mercury, such as methylmercury, during the experiment would allow us to estimate a toxicity potential based on environmental speciation.

## 5. Conclusions

The ecological restoration of alluvial mining sites in French Guiana is recent and essential for the sustainable development of the gold industry in tropical regions. In this study, the comparison of the soil quality of five vegetation restoration modalities, as well as a simulation of the hydromorphic period with an anaerobic mesocosms experiment, revealed very contrasting results on reducing microbial activities and on the dynamics of metallic elements. First, the results showed a difference in terms of physicochemical properties between restored and non-restored sites. Aca (*Acacia mangium*) and Mix (*Acacia mangium* and *Clitoria racemosa*) sites showed the positive effect of successful ecological restoration. Second, the simulation of flooding periods with the mesocosms showed interactions between iron- and sulfate-reducing activities depending on the rehabilitated sites. The two old restored sites (Aca and Mix) had predominant iron-reducing activities with the carbon and sulphur cycles coupled and a high potential for solubilization of iron and manganese. The restored Cli site and non-restored Lyc site had predominant sulfate-reducing activities and very low iron-reducing activities. Regarding the dynamics of mercury, its solubilization was twice as important at the two sites without tree cover restoration, which could lead to higher methylation potentials. Because these results provide information on the dynamics of microbial communities in association with the mobility of metallic elements, it seems necessary to further describe these results by characterizing the nature of metal oxides in the soils and estimating their environmental speciation according to the mode of ecological restoration.

## Figures and Tables

**Figure 1 microorganisms-09-01702-f001:**
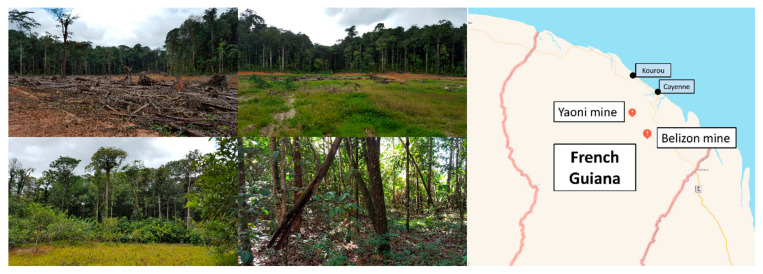
Illustration of some examples of different stages of ecological restoration on former Guyanese gold mining sites and geographical location of the Yaoni and Belizon mines in French Guiana. Non-restored site (**top left**), ecological restoration with herbaceous species after 3 years ((**top right**), Lyc), ecological restoration with fabaceous species after 3 years ((**bottom left**), Cli), ecological restoration with fabaceous species after 18 years ((**bottom right**), Mix). The pictures were taken during a second sampling campaign in April 2016 at sites undergoing ecological restoration.

**Figure 2 microorganisms-09-01702-f002:**
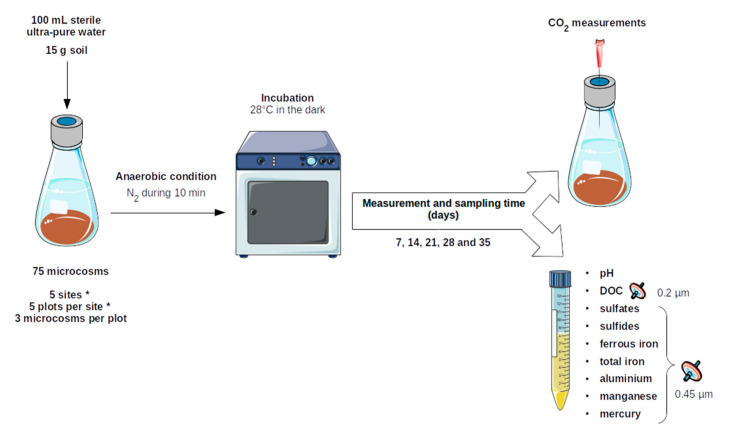
Description of the experimental set-up. First, preparation of the culture media in microcosms, then flushing with N_2_ to switch to anaerobic conditions. Incubation in the dark at 28 °C for 35 days. Before each sampling, CO_2_ measurements are carried out, then the culture medium is taken with a syringe, and the different parameters are measured.

**Figure 3 microorganisms-09-01702-f003:**
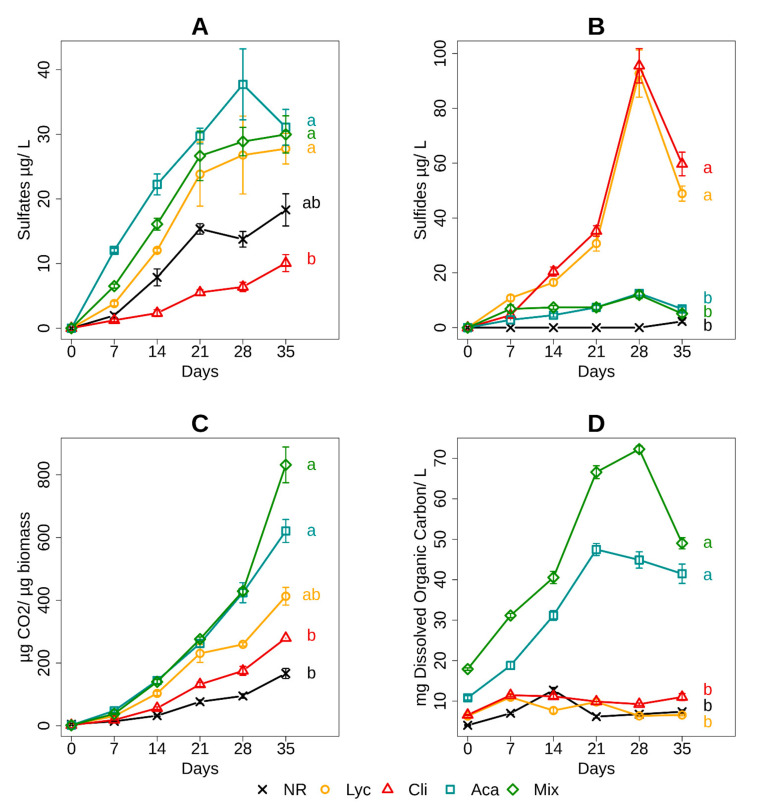
Sulfates (**A**) and sulfides (**B**) concentrations, microbial respiration (**C**) and dissolved organic carbon (**D**) during 35 days of anaerobic incubation in the five sites (*n* = 5, mean ± SD). For each site and at the end of each curve, an indicator letter has been added. Different letters (a,b,c,d) between sites indicate significant differences with *p* < 0.05 in the evolution of the measured parameter over time, obtained by repeated-measures ANOVA at the 5% critical threshold.

**Figure 4 microorganisms-09-01702-f004:**
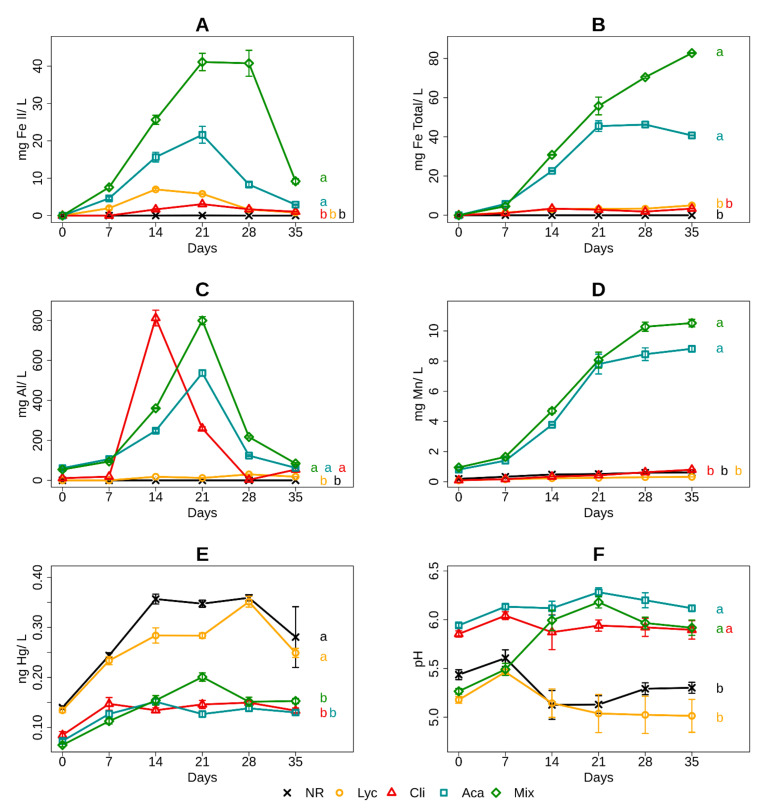
Fe(II) (**A**), total iron (**B**), Al (**C**), Mn (**D**), Hg (**E**) concentrations and pH (**F**) during 35 days of anaerobic incubation in the five sites (*n* = 5, mean ± SD). For each site and at the end of each curve, an indicator letter has been added. Different letters (a,b,c,d) between sites indicate significant differences with *p* < 0.05 in the evolution of the measured parameter over time, obtained by repeated-measures ANOVA at the 5% critical threshold.

**Figure 5 microorganisms-09-01702-f005:**
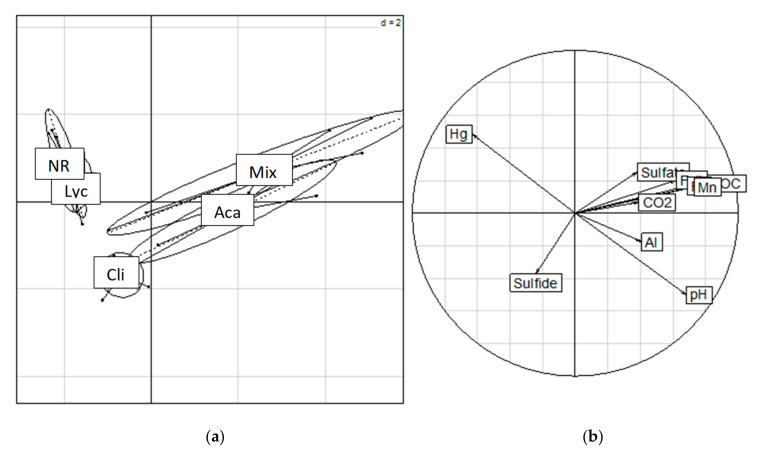
Principal component analysis (PCA) on the data monitored during the anaerobic experiment with types of vegetation cover. Left (**a**): projection of data set variability plotted on a factorial map of the first two discriminating axes according to respective Labels on the gravity center correspond to the types of vegetation cover: NR. Non-Restored with only spontaneous vegetation, Lyc. *Lycopodiella sp*, Cli. *C. racemosa*, Aca. *A. mangium*, Mix: Mixed fabaceous culture. To sub-factor of respective factor. Right (**b**): correlation circles plot with variable vectors (pH, CO_2_, DOC, SO_4_^2−^, S^2−^, Fe (II), Fe, Al, Mn, Hg) for each respective factor. Eigen values 54.4% and 17.5% for axes 1 and 2, respectively. Rand test: simulated *p*-value: 0.001. Explained variance: 47.1%.

**Table 1 microorganisms-09-01702-t001:** Ctot, Ntot, Ptot: Total C, N and P; TOC: Total Organic Carbon; MBC: Microbial biomass carbon; Total Fe, Al, Hg: Total Iron, Aluminum and Mercury content, (*n* = 5, mean ± SD). For each parameter, values followed by different letters (a,b,c,d) differ significantly with *p* < 0.05 with Tukey HSD test.

Sites						pH	Total						
	C_tot_	TOC	MBC	N_tot_	P_tot_		Fe	Mn	Al	Hg			
	g·kg^−1^	g·kg^−1^	mg·kg^−1^	g·kg^−1^	g·kg^−1^	pH-H_2_O	g·kg^−1^	g·kg^−1^	g·kg^−1^	µg·kg^−1^	Clay%	Silt%	Sand%
**NR**	6.01 ± 0.32 ^a^	3.88 ± 0.04 ^a^	223 ± 10.8 ^a^	0.40 ± 0.03 ^a^	0.21 ± 0.04 ^ab^	5.26 ± 0.23 ^b^	65.15 ± 3.17 ^b^	0.52 ± 0.05 ^c^	53.88 ± 1.53 ^c^	0.23 ± 0.01 ^a^	46.4 ± 3.6 ^b^	41.5 ± 2.8 ^c^	3.3 ± 0.2 ^a^
**Lyc**	9.55 ± 0.73 ^b^	5.84 ± 0.35 ^b^	385.9 ± 21.5 ^b^	0.45 ± 0.01 ^a^	0.27 ± 0.07 ^b^	4.66 ± 0.07 ^a^	44.56 ± 0.77 ^a^	0.10 ± 0.003 ^a^	41.33 ± 1.77 ^b^	0.23 ± 0.07 ^a^	22.2 ± 5.1 ^a^	65.8 ± 3.2 ^d^	11.8 ± 1.2 ^b^
**Cli**	7.01 ± 0.83 ^a^	5.0 ± 0.4 ^b^	844.4 ± 14.03 ^c^	0.40 ± 0.04 ^a^	0.17 ± 0.02 ^a^	5.0 ± 0.03 ^b^	113.3 ± 4.2 ^c^	0.11 ± 0.004 ^a^	25.5 ± 2.15 ^a^	0.28 ± 0.03 ^a^	21.3 ± 1.6 ^a^	24.8 ± 3.7 ^a^	53.9 ± 2.5 ^d^
**Aca**	14.11 ± 1.2 ^c^	10.55 ± 0.62 ^c^	944 ± 79 ^c^	1.12 ± 0.07 ^b^	0.37 ± 0.12 ^c^	4.21 ± 0.07 ^a^	53.69 ± 4.15 ^a^	0.25 ± 0.05 ^b^	45.31 ± 6.51 ^b^	0.48 ± 0.04 ^b^	18.4 ± 2.5 ^a^	35.6 ± 3.9 ^b^	41.6 ^a^ ± 4.4 ^d^
**Mix**	22.88 ± 1.5 ^d^	15.58 ± 1.41 ^d^	1103 ± 41 ^d^	1.81 ± 0.10 ^c^	0.76 ± 0.15 ^d^	4.66 ± 0.11 ^a^	71.02 ± 1.37 ^b^	0.54 ± 0.03 ^c^	43.76 ± 1.21 ^b^	0.41 ± 0.05 ^b^	19.8 ± 3.8 ^a^	32.4 ± 4.9 ^b^	36.4 ± 5.9 ^c^

**Table 2 microorganisms-09-01702-t002:** Effects of vegetation and incubation period in anaerobic conditions on soil pH, heterotrophic respiration, dissolved organic carbon, sulfates and sulfides content and iron, aluminum, manganese and mercury solubilization for five rehabilitated mining sites in mesocosm experiment in a two-way ANOVA and repeated-measures ANOVA. Values in bold indicate significant differences at the 0.05% critical level. An * means a highly significant result (*p* < 0.001).

		Sites	Time	Sites × Time	Repeated-Measures ANOVA
	Df	4	5	20	5
pH	F-value	**114 ***	0.9	**4.8 ***	0.61
Heterotrophic respiration (µg CO_2_·µg Bio C^−1^)		**175 ***	**513**	**35.6**	**14 ***
Dissolved Organic Carbon (mg DOC·L^−1^)		**215 ***	**108 ***	**105 ***	1.6
Sulfates (µg SO_4_^2−^·L^−1^)		**59.9 ***	**96.3 ***	**4.4 ***	**21 ***
Sulfides (µg S^2−^·L^−1^)		**305 ***	**214 ***	**58.5 ***	**3.7**
Soluble Fe (II) (mg Fe(II)·L^−1^)		**280 ***	**174 ***	**64.7 ***	**2.8**
Total soluble Fe (mg Fe·L^−1^)		**534 ***	**285 ***	**120 ***	**2.65**
Total soluble Al (mg Al·L^−1^)		**956 ***	**864 ***	**387 ***	**3.6**
Total soluble Mn (mg Mn·L^−1^)		**446 ***	**375 ***	**116 ***	**3.2**
Total soluble Hg (mg Hg·L^−1^)		**205 ***	**66.5 ***	**6.2 ***	**11.3 ***

**Table 3 microorganisms-09-01702-t003:** Pearson correlation coefficients for the data monitored during anaerobic experiment in the culture medium for all sampling points during the experiment. Values in bold indicate significant differences at the 0.05% critical level.

	pH	Sulfate	Sulfide	CO_2_	DOC	Fe(II)	Fe	Mn	Al	Hg
pH	**1**	0.232	−0.135	0.288	**0.603**	**0.470**	**0.551**	**0.595**	**0.479**	**−0.621**
Sulfate		**1**	0.141	**0.810**	**0.666**	**0.527**	**0.742**	**0.765**	0.229	0.180
Sulfide			**1**	0.197	−0.209	−0.117	−0.148	−0.186	−0.070	0.102
CO_2_				**1**	**0.609**	0.343	**0.800**	**0.786**	0.095	−0.049
DOC					**1**	**0.876**	**0.926**	**0.940**	**0.514**	−0.286
Fe(II)						**1**	**0.749**	**0.719**	**0.629**	−0.141
Fe							**1**	**0.977**	**0.409**	−0.218
Mn								**1**	**0.384**	−0.252
Al									**1**	−0.212
Hg										**1**

## Data Availability

Data are available by contacting the corresponding authors.

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
