# Peer review of "The Impact of Ecological Restoration on Biogeochemical Cycling and Mercury Mobilization in Anoxic Conditions on Former Mining Sites in French Guiana"

_microorganisms, 2021, doi:10.3390/microorganisms9081702_

Round 1
Reviewer 1 Report
- There are only a very small part of the results are related to the "bacterial" iron-reducing activity. Thus, in my opinion, this study is not fit for this journal that chiefly focusing on microorganisms. By contrast, most of the content is about the restoration ability of fabaceous plants.
- The layout of all tables is not satisfying.
- A significant difference between means should be applied in all data in the figures.
- Avoid using first-person writing, such as we, our, etc.
- In section 2.3.1. Bacterial metabolism, how do the authors know the sulphur oxidation and sulfate reduction were contributed by bacteria? If they are bacteria metabolism, what kind of bacteria were involved?
- The layout of the text is not satisfying, ex. subscript, space, etc.
- It needs iconic photos for the setup of microcosms.
- line 317: Although dissolution of - Although the dissolution of
- line 272: bioavailability of heavy metals - the bioavailability of heavy metals
- line 212: with this results - with these results
- line 207: characterize labile and recalcitrant organic matter - characterize the labile and recalcitrant organic matter
- Several paragraphs are too long and need to be separated into smaller paragraphs or reduced their size.
Author Response
Dear reviewer,
Thank you for your time in reviewing this article.
1 : There are only a very small part of the results are related to the "bacterial" iron-reducing activity. Thus, in my opinion, this study is not fit for this journal that chiefly focusing on microorganisms. By contrast, most of the content is about the restoration ability of fabaceous plants.
The results presented in this article provide information on biogeochemical cycles in the context of ecological restoration of mining sites. I fully appreciate that this article is relatively multidisciplinary and does not focus exclusively on microbial communities in tropical soils. Nevertheless, considering the keywords of this special issue "Critical Zone, biogeochemical cycling of elements, microbial communities, transportation and transformation of elements" I consider this article to be relevant for this special issue. Mining sites in the Amazon, often polluted with mercury, are critical zones for the transfer of metals from the soil into rivers and it is essential to understand the mechanisms of transfer and release of mercury. In this paper I focused on the release of metallic elements under anaerobic conditions to simulate the flooding periods of these mining sites undergoing ecological restoration. The results show that SRB and IRB activities seem to be influenced by the vegetation cover, and thus the presence of fabaceous species. In view of the results I thought it was important to link the observations to ecological restoration. Nevertheless, the overriding message is that there are differences in the functioning of biogeochemical cycles that are not necessarily correlated with microbial biomass, so there are different communities at mine sites, and it seems that ecological restoration with fabaceae limits the release of mercury. Since almost all mercury leaching mechanisms are biotic and related to SRB and IRB activities, the results presented in this article are, in my opinion, in line with the theme of the special issue.
2 : The layout of all tables is not satisfying.
I found that the tables were not satisfactory and that they did not correspond to the tables I uploaded. I have reformatted the tables.
3 : A significant difference between means should be applied in all data in the figures.
I have updated the soil properties table with the results of the anova tests. For the figures, I had previously described the significance of each of the variables through the repeated measures anova presented in Table 3. However, I did not include the results of the 2-to-2 pairings in the figures. I have therefore added the indicators of significance to the figures, which allows a more complete reading of the results.
4 : Avoid using first-person writing, such as we, our, etc.
I have corrected the first person forms to passive forms.
5 : In section 2.3.1. Bacterial metabolism, how do the authors know the sulphur oxidation and sulfate reduction were contributed by bacteria? If they are bacteria metabolism, what kind of bacteria were involved?
The reduction of sulphur is concomitant with the production of CO2 and therefore concomitant with anaerobic bacterial activity. This activity is also described in the literature as a driver of mercury methylation. Most of the time, the groups responsible for these activities are Desulfosporosinus sp, Desulfotomaculum sp, and Desulfovibrio sp, and Desulfobulbus sp, (the best known are Desulfovibrio vulgaris and as Desulfovibrio desulfuricans), but many genus are observed by the authors. In view of the results we probably have the same genus and molecular studies should be performed to confirm this.
Furthermore, some authors who have observed the same things as us in soils or sediments have succeeded in inhibiting this activity by using a molybdate-based inhibitor specific for sulphate-reducing bacteria. Also, in sterilised soils, which corresponds to control experiments, no activity of this type was observed, which shows that the reduction of sulphur is mainly of biotic origin.
6 : The layout of the text is not satisfying, ex. subscript, space, etc.
I have rearranged the text to remove the spaces. The layout is not perfect at this stage but I will check with the journal administration to refine the layout according to their criteria.
7 : It needs iconic photos for the setup of microcosms.
I have real pictures of the microcosms but the quality is not very high. Instead I have added a simplified diagram of the experimental set-up to summarise the steps involved in preparing the microcosms and measuring the experimental parameters.
8 : line 317: Although dissolution of - Although the dissolution of
I have corrected
9 : line 272: bioavailability of heavy metals - the bioavailability of heavy metals
I have corrected
10 : line 212: with this results - with these results
I have corrected
11 : line 207: characterize labile and recalcitrant organic matter - characterize the labile and recalcitrant organic matter
I have corrected
12 : Several paragraphs are too long and need to be separated into smaller paragraphs or reduced their size.
I have updated
Reviewer 2 Report
The above manuscript is dealing with the biogeochemical cycle of several metals that will influence mercury mobilisation and consequently its toxicity in mining sites ecologically restored from French Guiana territory. Hydromorphic conditions and the resulting anaerobic conditions that can promote IRB and SRB bacterial activities were also evaluated with incubation experiments, and the results obtained from these experiments will improve the knowledge of microbial communities at rehabilitated sites in Guyana and the ability to assess the effect of this rehabilitation on soil quality.
In general, the manuscript is well written, contain many and very useful data and will help to expand the knowledge of mercury dynamics in these type of environments. Some improvements should be done in the results sections, relatively to some figures that are referenced in the text but were not presented). The discussion is very well written, respond in a good way to the objectives of the study earlier presented, and, in my opinion, does not need to undergo significant changes after the suggested alterations to the results section.
I suggest the publication of this manuscript after a minor revision of some specific details along the text, indicated in the attached file as comments/suggestions and corrections.

Author Response
Dear reviewer,
Thank you for your time in reviewing this article.
I have fully taken into account the remarks you have sent me in the attached file. I will repeat some of the points in this reply.
This research article is indeed pioneering in Guyana. Some authors have characterised the communities of mining sites (Da Silva, Harris), the mercury cycle (Guedron) or the nitrogen cycle (Schimman), but on sites in the process of ecological restoration there is currently a huge lack of knowledge and very little hindsight. For this reason, my research work focused on the characterisation of biogeochemical cycles and microbial activities as a proxy to validate ecological restoration protocols. I have provided additional information in the introduction to highlight this information.
I have added the reference of the analyses of metals in soil samples. The principle of dissolution with aqua regia in microwave-treated Teflon tubes is the standard procedure for dissolving metals in soil phases. The addition of HF is often advisable if the soil contains a significant amount of mineral matter, which was the case for the samples. The full principles are described in the ISO standard, and are used by many authors. What differs from one article to another is often the quantity of soil used, as well as the volumes of acid. In my case, the quantities of soil and acid were calibrated according to the recommendations of the Teflon tube manufacturer, thus allowing the extraction to be optimised.
As for the results, I have updated the soil properties table with the results of the anova tests. For the figures, I had previously described the significance of each of the variables through the repeated measures anova presented in Table 3. However, I did not include the results of the 2-to-2 pairings in the figures. I have therefore added the indicators of significance to the figures, which allows a more complete reading of the results. Some results may appear slightly contradictory for pH and DOC. Table 3 shows that there is no significant difference overall between the samples, however the 2-to-2 matching tests showed differences between some samples. This result is however standard with multi-factor anova tests, and overall non-significance does not prevent some significant differences with 2-to-2 matching tests.
I have inserted missing references in the text and corrected typographical and formatting errors.
Reviewer 3 Report
The research is cutting edge. They were done methodically correctly. The methodology has been good described, but please write down when exactly, i.e. year and month, soil samples were taken for analysis and what were the weather conditions, e.g. temperature, humidity, etc.
The results obtained have been well documented in the Tables and Figures. In Figs. 2 and 3 you show the standard deviation (SD), but it would be nice to show homogeneous groups as well. That would improve the quality of the publication.
The conclusions are correctly constructed and correspond to the purpose of the manuscript.
Additionally, the authors in the introduction, the authors should emphasize the novelty of this study, the new hypothesis that they intended to demonstrate and its new goals based on the existing knowledge.
Make sure all items are cited in the text and in the table of contents and vice versa.
Author Response
Dear reviewer,
Thank you for your time in reviewing this article.
I have added a brief description of the sampling conditions. It was the rainy period in French Guiana and several days of fieldwork had been cancelled due to heavy rainfall making some of the forest "roads" impassable. Because of the high ambient humidity and the distance from the IRD laboratory in Cayenne, the samples were all pre-dried in the open air to avoid rotting, before being dried in controlled conditions in the laboratory.
Regarding the results, I have updated the soil properties table with the results of the anova tests. For the figures, I had previously described the significance of each of the variables through the repeated measures anova presented in Table 3. However, I did not include the results of the 2-to-2 pairings in the figures. I have therefore added the indicators of significance to the figures, which allows a more complete reading of the results. Some results may appear slightly contradictory for pH and DOC. Table 3 shows that there is no significant difference overall between the samples, however the 2-to-2 matching tests showed differences between some samples. This result is however standard with multi-factor anova tests, and overall non-significance does not prevent some significant differences with 2-to-2 matching tests.
I have inserted the missing references in the text.
This research article is indeed pioneering in Guyana. Some authors have characterised the communities of mining sites (Da Silva, Harris), the mercury cycle (Guedron) or the nitrogen cycle (Schimman), but on sites in the process of ecological restoration there is currently a huge lack of knowledge and very little hindsight. For this reason, my research work focused on the characterisation of biogeochemical cycles and microbial activities as a proxy to validate ecological restoration protocols. I have provided additional information in the introduction to highlight this information.
Round 2
Reviewer 1 Report
The manuscript has been revised accordingly.